# Analysis of a Combined HBHA and ESAT-6-Interferon-γ-Release Assay for the Diagnosis of Tuberculous Lymphadenopathies

**DOI:** 10.3390/jcm12062127

**Published:** 2023-03-08

**Authors:** Françoise Mascart, Maya Hites, Emmanuelle Watelet, Gil Verschelden, Christelle Meuris, Jean-Luc Doyen, Anne Van Praet, Audrey Godefroid, Emmanuelle Petit, Mahavir Singh, Camille Locht, Véronique Corbière

**Affiliations:** 1Laboratory of Vaccinology and Mucosal Immunity, Université Libre de Bruxelles (ULB), 1070 Brussels, Belgiumveronique.corbiere.ulb@gmail.com (V.C.); 2Clinic of Infectious and Tropical Diseases, Hôpital Universitaire de Bruxelles (HUB)-Hôpital Erasme, Université Libre de Bruxelles (ULB), 1070 Brussels, Belgium; 3Department of Pneumology, Clinique St-Anne/St-Remi—Chirec, 1070 Brussels, Belgium; 4Department of Internal Medicine, Universitair Ziekenhuis Brussel—UZ Brussel, Vrije Universiteit Brussel (VUB), 1090 Brussels, Belgium; 5Department of Infectious Diseases, Liège University Hospital, 4000 Liège, Belgium; 6U-1019—UMR8204, Center for Infection and Immunity of Lille (CIIL), CNRS, Inserm, CHU Lille, Institut Pasteur de Lille, University of Lille, 59000 Lille, France; 7Lionex Diagnostics and Therapeutics, 38126 Braunschweig, Germany

**Keywords:** tuberculous lymphadenopathy, IGRA, HBHA, ESAT-6

## Abstract

Background and Objectives: The incidence of tuberculosis lymphadenopathy (TBLA) is increasing, and diagnostic procedures lack sensitivity and are often highly invasive. TBLA may be asymptomatic, and differential diagnosis with other adenopathies (ADPs) is difficult. We evaluated a blood-cell interferon-γ release assay (IGRA) with two different stage-specific mycobacterial antigens for the differential diagnosis of ADP suspected of mycobacterial origin. Methods: Twenty-one patients were included and divided into three groups: (1) cervical/axillar ADP (*n* = 8), (2) mediastinal ADP (*n* = 10), and (3) disseminated ADP (*n* = 3). The mycobacterial antigens used for the IGRA were the heparin-binding haemagglutinin (HBHA) and the early-secreted antigenic target-6 (ESAT-6), a latency-associated antigen and a bacterial replication-related antigen, respectively. Diagnosis of TBLA based on microbiological results and/or response to anti-TB treatment was obtained for 15 patients. Results: An IGRA profile highly suggestive of active TB (higher IFN-γ response to ESAT-6 compared to HBHA) was found for 3/6 TBLA patients from group 1, and for all the TBLA patients from groups 2 and 3, whereas this profile was not noticed in patients with a final alternative diagnosis. Conclusion: These results highlight the potential value of this combined HBHA/ESAT-6 IGRA as a triage test for the differential diagnosis of ADP.

## 1. Introduction

In contrast to the reduction in the total number of tuberculosis (TB) cases observed before the COVID-19 pandemic, an increase in extra-pulmonary TB (EPTB) has been reported. Twenty percent of TB cases present with EPTB, and among them, tuberculous lymphadenopathy (TBLA) is the most common form, reaching 50% of all EPTB cases [1,2,3]. TBLA may occur almost anywhere in the body and can sometimes develop in the absence of pulmonary involvement. Cervical localization represents 63% of TBLA. However, even if adenopathy (ADP) can be easily observed, the diagnosis of TBLA is difficult due to its pauci-bacillary nature. Diagnosis of mediastinal or abdominal TBLA is even more challenging, especially in the absence of lung involvement, as clinical signs and symptoms lack specificity and are sometimes absent. Computed tomography (CT) is a method of choice for the detection of mediastinal or abdominal ADP, but abnormalities are also not specific [4]. Only a positive *Mycobacterium tuberculosis* culture or PCR on biopsies, obtained by invasive procedures, allows for TBLA diagnosis confirmation. However, these approaches lack sensitivity [5].

Therefore, alternative approaches providing additional arguments in favor of TB may be of value before invasive procedures are considered. Blood-based interferon-γ-release assays (IGRA), such as the QuantiFERON (QFT), may be helpful, but positive QFT results are not able to discriminate latent TB infection (LTBI) from active TB. Consequently, they cannot be used to differentiate TBLA from LTBI subjects potentially suffering from sarcoidosis or lymphoma [6]. In contrast, IGRA using two different stage-specific mycobacterial antigens, heparin-binding haemagglutinin (HBHA) and early-secreted antigenic target-6 (ESAT-6), are able to differentiate LTBI from active TB [7,8,9,10,11]. Therefore, we retrospectively evaluated here the potential of this combined HBHA/ESAT-6 IGRA for the diagnosis of TBLA.

## 2. Materials and Methods

### 2.1. Study Protocol

Between 2016 and 2020, clinicians from Belgian hospitals requested a validated in-house HBHA/ESAT-6 IGRA [7,9] for suspected TBLA or for LTBI screening. The IGRA results helped clinicians to continue or stop investigations on TBLA. The final diagnosis of TBLA was based on microbiological proof of *M. tuberculosis* infection, or on response to treatment. This study analyzed retrospectively the accuracy of the combined HBHA/ESAT-6 IGRA to increase the diagnostic probability of TBLA in 21 patients suspected to present TBLA without pulmonary involvement.

### 2.2. Ethical Approval

All but three patients were included in a study on biomarker discovery in human TB, approved by the Ethics Committee ULB-Hôpital Erasme (OMO21, protocol P2016/252), and gave their written informed consent. For the three remaining patients, the “Comité d’Ethique Hospitalo-Facultaire Universitaire de Liège (707, protocol P2007/175)” approved the retrospective analysis of the clinical data in the context of evaluation of an IGRA for the diagnosis of TBLA.

### 2.3. Combined HBHA/ESAT-6 Interferon-γ-Release Assays

Validated HBHA/ESAT-6 IGRAs were performed as described [10]. Briefly, fresh peripheral blood mononuclear cells (PBMC) were suspended at 2 × 10^6^/mL in culture medium (RPMI 1640 medium supplemented with 40 μg/mL gentamicin, 50 μM β-mercaptoethanol, 1× non-essential amino acids, 1 mM sodium pyruvate, 2 mM glutamine, and 10% fetal calf serum) in the presence of IL-7 (1 ng/mL, R&D, Bio-Techne, UK) and in the presence or absence of HBHA (2 μg/mL) or ESAT-6 (5 μg/mL). Native HBHA was purified from *Mycobacterium bovis* BCG (strain 1173P2; World Health Organization) as previously described [12], and recombinant ESAT-6 was provided by Lionex (Braunschweig, Germany). After 24 h at 37 °C with 5% CO_2,_ cell supernatants were collected and frozen at −20 °C until IFN-γ measurement by enzyme-linked immunosorbent assays (IFN-γ Cytoset, Life Technologies, Ghent, Belgium) [7,10]. As negative and positive controls, IFN-γ concentrations were measured in the cell supernatants in absence of any added stimulant and in presence of 0.5 μg/mL staphylococcal enterotoxin B (SEB; Sigma-Aldrich, Bornem, Belgium), respectively. Control values <50 pg/mL in the negative control and >200 pg/mL in response to SEB were required.

The commercially available QFT (Qiagen), only requested by clinicians for a few patients, was performed following the manufacturer’s instructions.

## 3. Results

### 3.1. Main Clinical and Demographic Data

The patients were classified into three different groups, according to the main localization of the ADP. Group 1 comprised eight patients who initially presented with a cervical and/or axillar ADP obviously discovered by the patient (Table 1). Group 2 consisted of ten patients in which mediastinal ADPs were discovered. These patients had a chest radiograph or CT together with a blood sampling for HBHA/ESAT-6 IGRA in the context of (1) follow-up post lymphoma, smoking, or screening for latent TB in asymptomatic patients (Table 2, n°9–12); (2) cutaneous or ocular lesions (Table 2, n°13–16); and (3) cough and dyspnea (Table 2, n°17–18). An ^18^F-fluorodeoxyglucose positron emission tomography (PET Scan) was performed in the case of an IGRA result that was highly suggestive of active TB despite negative chest imagery. Finally, group 3 comprised three patients with disseminated TBLA or miliary TB who initially presented with general symptoms of unknown origin. A blood sampling was performed along with a PET Scan as part of the investigations or secondary to the IGRA results (Table 3, n°19–21).

The main clinical and demographic data are summarized in Table 1, Table 2 and Table 3. The median age of all patients was 45 years (range: 18–79 years).

### 3.2. Diagnosis in Group 1: Cervical and/or Axillar Adenopathy

Six patients presented with a unilateral cervical ADP (Table 1, n°1 to n°6), one (n°7) with a bilateral cervical ADP and a unilateral fistulized axillar ADP, and one (n°8) with a suppurative axillar ADP (Table 1). None of these patients had pulmonary involvement as demonstrated by chest radiograph and/or CT, but mediastinal ADPs were discovered in patient n°7. Results from histopathological examination of ADP biopsies were non-specific, and were highly suggestive of TB only in two patients (necrotic granuloma) (Table 1, n°1 and n°3). These biopsies yielded positive microbiological results for *M. tuberculosis* in all patients (positive PCR, including a positive culture for 3/6), except for n°5 and n°8. As patient n°6 mentioned abdominal discomfort and perspiration, an abdominal CT was performed. Results were highly suggestive of peritoneal TB leading to a diagnosis of miliary TB. All the other patients, except n°5 and n°8, were diagnosed as TBLA (Figure 1). The six TB patients received appropriate quadritherapy, which resulted in clinical improvement.

IGRA results were strongly positive for the six patients with TBLA, with isolated IFN-γ secretion to ESAT-6 for one of them (patient n°7) and higher IFN-γ secretion in response to ESAT-6 than to HBHA for two others (patients n°3 and n°6) (Table 4 and Figure 1). These IGRA profiles, which are highly suggestive of active TB [7,9,11,13], characterized patients with a unilateral cervical ADP (n°3), a miliary TB (n°6), and bilateral cervical ADPs with axillar ADP (n°7). The other patients with unilateral cervical ADP had lower IFN-γ secretion in response to ESAT-6 than to HBHA, a profile characteristic of patients who better control *M. tuberculosis* infection [9,14,15]. IGRA results were negative for patient n°8 with a final diagnosis of leukemia, and were suggestive of latent TB (isolated low IFN-γ response to HBHA) for patient n°5 with a presumed diagnosis of IgG4-related disease (Table 2).

**Table 1 jcm-12-02127-t001:** Characteristics of patients included in group 1 with cervical/axillar adenopathies.

Patient (Number, Category)	Clinical Manifestation	Gender	Age (y)	Country of Origin	Risk Factor	Imagery		Biopsy	*Mtb* Detection		CRT	Final Diagnosis
						Chest X-ray/CT	PET Scan (Positivity)	Histology	PCR	Cult		
1	Unilateral ADP	F	29	Morocco	Pregnant	Nl	ND	Necrotic granuloma	Pos	Neg	Pos	Lymph node TB
2	Unilateral ADP	M	49	Morocco	/	Nl	ND	Granuloma	Pos	Neg	Pos	Lymph node TB
3	Unilateral ADP	F	20	Guinea	/	Nl	ND	Necrotic granuloma	Pos	Pos	Pos	Lymph node TB
4	Unilateral ADP	M	52	Morocco	/	Nl	ND	Necrosis	Pos	Pos	Pos	Lymph node TB
5	Unilateral ADP	M	18	The Netherlands	TB contact 3 y earlier	Nl	Submandi-bular ADP	Lympho-plasmocytair infiltrates	Neg	Neg	NA	IgG4-related disease (suspicion)
6	Unilateral ADP, abdominal pain, sweat	F	30	Morocco	3 months post-partum	Nl	ND	Necrosis	Pos	Neg	Pos	Miliary TB
7	Bilateral ADPs/fistulized axillar ADP	F	18	Congo	TB contact	Mediastinal ADPs	ND	ND	Pos	Pos	Pos	Disseminated lymph node TB
8	Suppurative axillar ADP	M	65	Belgium	Travel in endemic countries	Axillar ADP	ND	ND	Neg	Neg	NA	Leukemia

*Mtb*: *Mycobacterium tuberculosis*; ADP: adenopathy; F: Female; M: Male; Nl: normal; ND: not done; NA: Not applicable; LTBI: latently TB-infected; Pos: positive; Neg: negative; CRT: clinical response to treatment; /: no risk factor; grey lines: patients with no TB.

**Table 2 jcm-12-02127-t002:** Characteristics of patients included in group 2 with mediastinal adenopathies.

Patient (Number, Category)	Clinical Manifestation	Gender	Age (y)	Country of Origin	Risk Factor	Imagery		Biopsy	*Mtb* Detection		CRT	Final Diagnosis
						Chest X-ray/CT	PET Scan (Positivity)	Histology	PCR	Cult		
	**Asymptomatic**											
9	Lymphoma follow-up	M	79	Morocco	Lymphoma > 10 y	Mediastinal ADPs	ND	Epitheloïd granuloma	Neg	Neg	ND	Lymph node TB
10	Smoking	M	57	Greece	Diabetes, cardiopathy	Mediastinal ADPs	Mediastinal ADPs	NA	ND	ND	NA	Sarcoidosis (suspicion)
11	LTBI screening before anti-TNF-α	M	64	Morocco	Rheumatoid arthritis	Small pulmonary nodules	Axillar and mediastinal ADPs	Epitheloïd cells	Neg	Neg	Pos	Lymph node TB
12	LTBI screening in a healthcare worker	F	45	Congo	TB contact and past-TB	Nl	Mediastinal ADPs	Epitheloïd granuloma	Neg	Neg	Pos	Lymph node TB
	**Cutaneous/Ocular lesions**											
13	Erythema nodosum, polyarthralgia, cutaneous inflammatory lesion on the forearm, sub-clavicular ADP	F	36	Morocco	Past-treated latent TB	Mediastinal ADPs; one pulmonary nodule	ND	Necrotic granuloma (cutaneous biopsy)	Pos	Pos	Pos	Cutaneous TB and disseminated lymph node TB
14	Erythema nodosum	M	29	Guinea	/	Mediastinal ADPs	ND	Granuloma	Pos	Pos	Pos	Lymph node TB
15	Erythema nodosum	F	33	Brazil	/	Mediastinal ADPs	ND	Granuloma	Neg	Neg	NA	Sarcoidosis
16	Granulomatous uveitis	F	46	Morocco	/	Nl	Cervical and mediastinal ADPs	Necrotic granuloma	Neg	Neg	Pos	Lymph node TB
	**Cough/Dyspnea**											
17		M	62	Kosovo	/	Mediastinal ADPs	Mediastinal ADPs	Nl	Neg	Neg	NA	Pancreatic neoplasia
18		F	65	Kosovo	/	Mediastinal ADPs	Axillar and mediastinal ADPs; mammary gland	ND	ND	ND	NA	Breast neoplasia

*Mtb*: *Mycobacterium tuberculosis*; ADP: adenopathy; F: Female; M: Male; Nl: normal; ND: not done; NA: Not applicable; LTBI: latently TB-infected; Pos: positive; Neg: negative; CRT: clinical response to treatment; /: no risk factor; grey lines: patients with no TB.

**Table 3 jcm-12-02127-t003:** Characteristics of patients included in group 3 with disseminated adenopathies.

Patient (Number, Category)	Clinical Manifestation General Symptoms	Gender	Age (y)	Country of Origin	Risk Factor	Imagery		Biopsy	*Mtb* Detection		CRT	Final Diagnosis
						Chest X-ray/CT	PET Scan (Positivity)	Histology	PCR	Cult		
19	Sweat, weight loss, vomiting	F	25	Italy	/	Nl	Cervical, mediastinal, liver hilar, and retroperitoneal ADP	Necrotic granuloma (liver hilar ADP)	Pos	Pos	Pos	Disseminated lymph node TB
20	Sweat, weight loss, cough	M	23	Erytree	/	Bilateral micro-nodules; mediastinal ADPs	Cervical, mediastinal, mesenteric ADPs, and liver	Necrotic granuloma (liver biopsy)	Pos	Pos	Pos	Miliary TB
21	Apathy, poor general condition	M	63	Belgium	/	Pulmonary nodules, mediastinal ADPs	Cervical, mediastinal, mesenteric, and liver hilar ADPs	Necrotic granuloma (liver hilar ADP)	Pos	Neg	Pos	Disseminated lymph node TB

*Mtb*: *Mycobacterium tuberculosis*; ADP: adenopathy; F: Female; M: Male; Nl: normal; ND: not done; NA: Not applicable; LTBI: latently TB-infected; Pos: positive; Neg: negative; CRT: clinical response to treatment; /: no risk factor.

**Figure 1 jcm-12-02127-f001:**
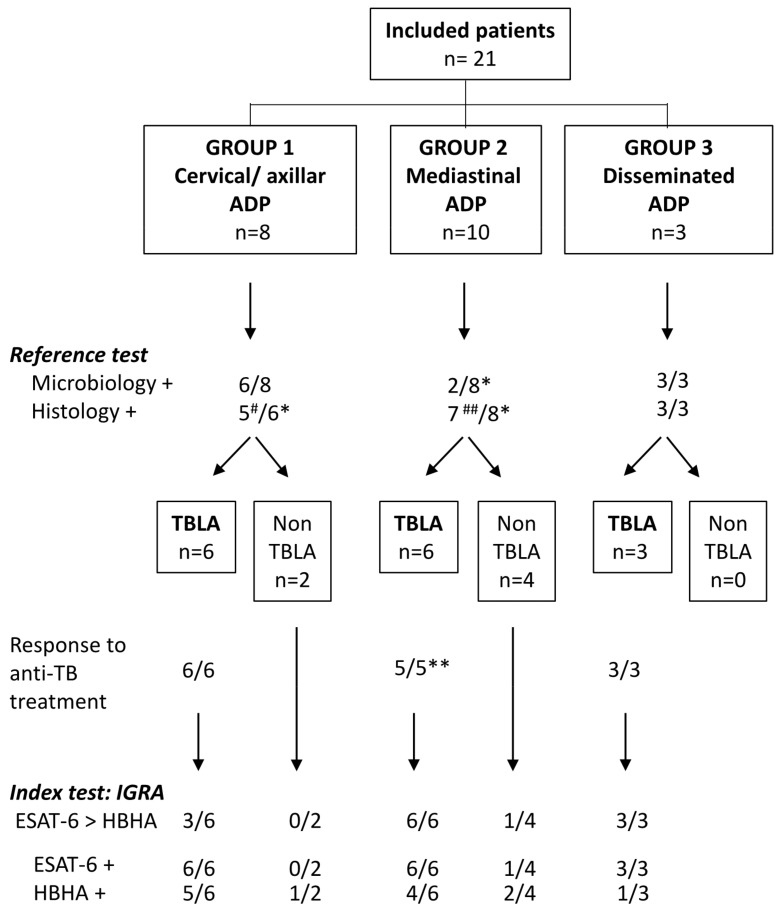
Flow chart comparing the index test to the reference test for the diagnosis of TBLA. Final diagnosis of TBLA among the 21 included patients was based on the reference test, the *M. tuberculosis* culture, and PCR performed on biopsies (summarized as microbiology in the figure), and in case of negativity, on results from histological analysis who were considered as helpful, albeit often not specific. Histology was considered as positive when necrotic granuloma was noticed and doubtful in the case of epitheloïd granuloma, granuloma without necrosis, or necrosis without granuloma. Diagnosis was confirmed by clinical and radiological responses to treatment. The index test was the combined HBHA/ESAT-6 IGRA and more precisely a higher IFN-γ response to ESAT-6 than to HBHA. ^#^: doubtful for 3 patients; ^##^: doubtful for 5 patients; *: not available for 2 patients; **: not available for 1 patient.

**Table 4 jcm-12-02127-t004:** IGRA results.

Patient (Number, Category)	*Mtb* Detection		HBHA (pg/mL)	ESAT-6 (pg/mL)	QFT: TB-1 (IU/mL)	QFT: TB-2 (IU/mL)
	PCR	Cult	50 pg/mL *	50 pg/mL *	0.35 IU/mL *	0.35 IU/mL *
**Group 1** **Cervical/Axillar ADP**						
1	Pos	Neg	3119	471	ND	ND
2	Pos	Neg	8257	4371	ND	ND
3	Pos	Pos	52	586	ND	ND
4	Pos	Pos	7967	1094	ND	ND
5	Neg	Neg	57	<10	ND	ND
6	Pos	Neg	1028	4616	2.54	ND
7	Pos	Pos	<10	582	>10	>10
8	Neg	Neg	<10	<10	0.07	0.09
**Group 2** **Mediastinal ADP**						

9	Neg	Neg	3711	4826	ND	ND
10	ND	ND	403	<10	0.00	0.00
11	Neg	Neg	139	4210	5.46	4.07
12	Neg	Neg	>9000	>9000	Indeterminate	Indeterminate

13	Pos	Pos	44	68	0.75	ND
14	Pos	Pos	4678	12,537	ND	ND
15	Neg	Neg	38	108	0.29	0.16
16	Neg	Neg	19	452	>10	>10

17	Neg	Neg	75	12	0.70	0.77
18	ND	ND	21	<10	0.00	0.00
**Group 3** **Disseminated ADP**						
19	Pos	Pos	32	670	ND	ND
20	Pos	Pos	87	831	>10	4.24
21	Pos	Neg	<10	774	7.42	>10

* positivity limit; ADP: adenopathy; ND: not done; Ind: indeterminate; grey lines: patients with no TB.

### 3.3. Diagnosis in Group 2: Mediastinal Adenopathy

Four patients were asymptomatic (Table 2, n°9–n°12) and mediastinal ADPs were discovered for two of them by thoracic CT. The two other subjects who were screened for LTBI (Table 2, n°11 and n°12) had no obvious chest CT abnormalities. Patient n°11 had a positive QFT, indicating infection with *M. tuberculosis*, whereas the QFT result in patient n°12 was indeterminate (elevated negative control) (Table 4). However, the results from the combined HBHA/ESAT-6 IGRA were highly suggestive of active TB for the latter two patients, with undetectable IFN-γ in the negative control sample even for patient n°12 (Table 4). Therefore, a PET Scan was performed and revealed mediastinal ADPs (Table 2). Lymph node biopsies were obtained by endoscopic ultrasound-guided fine needle aspiration (EBUS) for three of these asymptomatic patients and was contra-indicated for patient n°10 who was under anticoagulant therapy. For patients n°9, 11, and 12, histological examination provided non-specific abnormalities, and *M. tuberculosis* culture and PCR of lymph node biopsies were negative. Based on the results from the combined HBHA/ESAT-6 IGRA that were highly suggestive of active TB in patients n°9, 11, and 12 (higher IFN-γ responses to ESAT-6 than to HBHA) (Table 4), anti-TB treatment was initiated for these patients. Regression of the ADPs was noticed for patients n°11 and n°12, confirming the diagnosis of subclinical active TB. No follow-up was possible for patient n°9, who had left the country, so that the final diagnosis remained doubtful. In contrast, patient n°10 with no available biopsy had a positive IGRA to HBHA and no response to ESAT-6, a pattern characteristic of latent TB. The diagnosis was possible sarcoidosis in an LTBI subject.

Mediastinal ADPs were discovered by a thoracic CT in three patients presenting initially with cutaneous lesions (Table 2, n°13, 14, 15). Lymph node biopsies were obtained by EBUS for patients n°14 and n°15, and a biopsy of the cutaneous lesion was performed for patient n°13. Histological examination of all three revealed the presence of granulomas, which were necrotic only within the cutaneous biopsy of patient n°13. Only patients n°13 and n°14 had positive *M. tuberculosis* PCR and culture and were diagnosed as TBLA. They both had positive combined HBHA/ESAT-6 IGRA, with a limited response to ESAT-6 in patient n°13, and a higher response to ESAT-6 than to HBHA in patient n°14 (Table 4). Patient n°15 with negative microbiology had a high plasma concentration of angiotensin convertase (81.0 UECA; normal values: 12.0–68.0). As the IFN-γ response to ESAT-6 was low, albeit higher than the response to HBHA, which was not significant, the diagnosis of sarcoidosis in an LTBI patient was retained (Table 4). Patient n°16, with granulomatous uveitis, had a normal thoracic CT, and mediastinal ADPs were only detected by PET Scan, performed following strongly indicative HBHA/ESAT-6 IGRA results (Table 4). Histology of the lymph node biopsy was highly suggestive of TBLA (necrotic granuloma), while *M. tuberculosis* PCR and culture were negative. The diagnosis was confirmed by clear improvement of both the uveitis and the ADPs under TB treatment (Table 2).

Finally, mediastinal ADPs were discovered by thoracic CT in two patients presenting with cough and dyspnea (Table 2, n°17, n°18). The PET Scan confirmed hyper-metabolic ADPs in both patients, but microbiological results remained negative. Patient n°17 had a slightly positive QFT and a low response to HBHA indicating latent TB (Table 4). In this patient, pancreatic neoplasia was discovered. The QFT and the HBHA/ESAT-6 IGRA were negative for patient n°18 in whom breast neoplasia was diagnosed.

### 3.4. Diagnosis in Group 3: Disseminated Lymph Node TB

Three patients presenting with general symptoms of weight loss, apathy, and/or perspiration were diagnosed with disseminated TBLA or miliary TB (Table 3, n°19, n°20, n°21). A PET Scan performed in view of symptoms for patients n°19 and n°20, and secondary to the HBHA/ESAT-6 IGRA results, which was highly suggestive of TB for patient n°21 (Table 4), showed hypermetabolic cervical and mediastinal ADPs for all three, liver hilar ADPs for patients n°19 and n°21, and liver hypermetabolic activity for patient n°20 (Table 3). Biopsies were obtained from liver hilar ADP for patients n°19 and n°21, and from the liver for patient n°20. All three patients had positive histological (necrotic granulomas) and microbiological results on biopsies confirming a diagnosis of disseminated TBLA for patients n°19 and n°21, and of miliary TB for patient n°20 (Table 3). All three patients had positive HBHA/ESAT-6 IGRA with higher IFN-γ responses to ESAT-6 than to HBHA, suggestive of active TB (Table 4). A positive QFT was available for two patients, suggestive of an infection with *M. tuberculosis,* but it did not inform about active TB versus LTBI (Table 4). Clinical improvement was observed after initiating appropriate treatment confirming the diagnosis.

## 4. Discussion

Although its importance is largely underestimated [2,3], EPTB contributes significantly to TB-related morbidity and mortality, and TBLA is the most common form of EPTB in Europe and the USA. Diagnosis is difficult even in the presence of apparent cervical ADP, due to several possible differential diagnoses and the pauci-bacillary nature of TBLA. For mediastinal and/or abdominal ADP, which may be life-threatening, diagnosis is even more complicated, often delayed, and requires usually highly invasive procedures, combined with expensive imaging technologies, which are not universally available. This retrospective analysis of 21 patients with ADP examined the diagnostic potential of a non-invasive, combined blood-based HBHA/ESAT-6 IGRA for TBLA.

ESAT-6 IGRA results were positive for the six patients with cervical and/or axillar TBLA, whereas they were negative for the two patients with cervical/axillar ADP with a final alternative diagnosis (Figure 1, group 1). The IFN-γ response to ESAT-6 was higher than to HBHA in the two patients with disseminated TBLA or miliary TB, and in one patient with an isolated cervical ADP, whereas it was lower in response to ESAT-6 than to HBHA for three patients with isolated cervical ADP. We hypothesize that this latter profile, reported to be associated with effective control of the *M. tuberculosis* infection [9,14], may reflect a self-healing primary *M. tuberculosis* infection, during which isolated cervical ADP often develops [3]. In the mediastinal TBLA group (group 2), positive IGRA was obtained for the six patients, but a low positive response to ESAT-6 was also noticed for one patient classified as non-TB (Figure 1). It is of note that, except for one patient, mediastinal TBLA was characterized by very high IFN-γ responses to ESAT-6, associated to positive responses to HBHA, albeit lower than those to ESAT-6 (Table 4). Such IGRA profiles were previously reported to be highly suggestive of active TB, whereas the reverse profile (high response to HBHA with low or absent response to ESAT-6) is suggestive of LTBI [7,8,9,10,11,13,14,15]. Finally, the HBHA/ESAT-6 IGRA was positive for the three patients with disseminated TBLA (group 3) with a profile highly suggestive of active TB (Figure 1).

Comparing the IFN-γ responses to two different stage-specific mycobacterial antigens is a major advantage of the combined HBHA/ESAT-6 IGRA over the commercialized QFT, which cannot discriminate LTBI from active TB [6]. For three patients with mediastinal TBLA, the ADPs were only discovered by PET Scan performed secondary to the IGRA results, whereas the thoracic CT was normal. Similarly, for patient n°21 presenting with non-specific symptoms, disseminated ADPs were discovered by PET Scan performed secondary to the IGRA results. For all four patients, QFT was performed but did not suggest active TB. We show, thus, here that the comparison of IFN-γ responses to HBHA and ESAT-6 may be useful as a first step in the diagnosis of mediastinal and disseminated TBLA, as it may be helpful in the differential diagnosis of cervical APD. Results of this IGRA are available within 48 h and can therefore be used easily as a triage approach before engaging in sophisticated and invasive procedures. Negative IGRA to both antigens, negative ESAT-6-IGRA with positive HBHA-IGRA, or positive IGRA to both antigens with a lower IFN-γ response to ESAT-6 than to HBHA is not suggestive of active TB. In contrast, an isolated IFN-γ response to ESAT-6, a higher IFN-γ response to ESAT-6 than to HBHA, or an extremely high response to both antigens is highly suggestive of active TB. However, a higher IFN-γ response to ESAT-6 than to HBHA may also be associated with quiescent bacterial replication in subjects with latent TB; hence, in subjects more at risk of developing active TB [9,14]. Such results, especially in cases of low positivity, should therefore always be interpreted within the clinical context and in association with other blood results, such as those of angiotensin convertase plasma levels in case of differential diagnosis with sarcoidosis.

The limitations of this study are the low number of patients included with a TBLA in a low TB incidence country and the retrospective nature of the study. Based on these results, larger prospective studies should be performed to confirm the interest of the combined HBHA/ESAT-6 IGRA to increase the diagnostic suspicion of TBLA and to differentiate it from LTBI. Such studies would be especially useful in migrant populations who are characterized by a high frequency of TBLA [16]. In the present study, 12/15 patients with TBLA originated from Africa. TBLA in migrants has often an insidious onset, resulting from the reactivation of LTBI acquired before arrival in the host country. In this study, the six patients with mediastinal TBLA may be considered as subclinical TB, as three of them were strictly asymptomatic, and the three others presented non-specific symptoms (erythema nodosum, uveitis). Neither tuberculin skin tests nor commercial IGRAs are helpful in differentiating subclinical TB from LTBI [6]. The HBHA/ESAT-6 IGRA may, thus, be particularly useful for the detection of TBLA in migrants and its differential diagnosis from LTBI, allowing them to be provided with the appropriate antibiotic treatment.

## Data Availability

Data are unavailable due to ethical restrictions, but they will be made available on specific request.

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
