# Peer review of "Analysis of a Combined HBHA and ESAT-6-Interferon-γ-Release Assay for the Diagnosis of Tuberculous Lymphadenopathies"

_jcm, 2023, doi:10.3390/jcm12062127_

Round 1

Reviewer 1 Report

This original article provides interesting data to improve the diagnosis of tuberculosis lymphadenopathy. Results obtained are presented transparently and clearly, and provide a proof of concept for testing combined HBHA-ESAT-6 IGRA on larger prospective cohorts.

Minor comments:

Title: The results of the combined HBHA/ESAT-6 IGRA helped guide the diagnosis in some cases in this study, so the term "retrospective analysis" is not the most appropriate

Lines 36-37: According to Table 2 and lines 195-199, among the 15 patients diagnosed with TBLA, 1 patient, the patient 9, had negative microbiological result and he was lost to follow-up, making it unclear whether he responded to treatment. His diagnosis of TBLA was based on combined IGRA results. Accordingly, please clarify that the TBLA diagnosis was also based on combined IGRA result.

Lines 38-41: According to Table 4, the result was indeterminate for 1 TBLA patient 12, and a higher IFN-g response to ESAT-6 compared to HBHA was observed for patient 15 diagnosed for sarcoidosis. Please clarify these points.

Line 96: Please specify concentrations of HBHA and ESAT-6 used.

Line 124: According to Tables 1-3, age range was 18-79 years and not 18-66 years. Please correct.

Figure 1: Please specify in the foot note for group 2 that IGRA ESAT-6>HBHA result was indeterminate for 1 TBLA patient. Regarding the group 3, HBHA was negative for 2 patients (19 and 21) according to Table 4. Finally, it could be interesting to add statistical analysis to emphasize the added value of combined HBHA/ESAT-6 IGRA compared to IGRA tests alone.

Lines 167-169: Please specify the group.

Lines 172-174 and 266-267: The IGRA result ESAT-6<HBHA is indicated as characteristic of better control of Mtb infection, but according to the literature it also could be interpreted as the early stage of TB reactivation. Please clarify.

Table 4: Instead of the column “clinical manifestation” it could be more informative to indicate the microbiological results to avoid redundancy with Tables 1-3. Please at least justify this choice.

Lines 186-189: Please clarify why you consider HBHA/ESAT-6 IGRA result of the patient 12 as “highly suggestive of active TB”, despite the indeterminate result due to the elevated negative control.

Discussion: As stated for the title, the results of the combined HBHA/ESAT-6 IGRA helped guide the diagnosis in some cases in this study, so the term "retrospective analysis" is not the most appropriate.

Author Response

Title: The results of the combined HBHA/ESAT-6 IGRA helped guide the diagnosis in some cases in this study, so the term "retrospective analysis" is not the most appropriate.

Response: We agree with the reviewer’s comment and have deleted the term “retrospective” from the title and in the abstract.

Lines 36-37: According to Table 2 and lines 195-199, among the 15 patients diagnosed with TBLA, 1 patient, the patient 9, had negative microbiological result and he was lost to follow-up, making it unclear whether he responded to treatment. His diagnosis of TBLA was based on combined IGRA results. Accordingly, please clarify that the TBLA diagnosis was also based on combined IGRA result.

Response: Results from the combined IGRA were highly suggestive of active TB, and based on these results (line 216), anti-TB treatment was started. As no follow-up was possible, we were unable to confirm the diagnosis. We therefore add now that the final diagnosis for this patient remained doubtful (lines 219-220).

Lines 38-41: According to Table 4, the result was indeterminate for 1 TBLA patient 12, and a higher IFN-g response to ESAT-6 compared to HBHA was observed for patient 15 diagnosed for sarcoidosis. Please clarify these points.

Response:

  • Patient 12 had an indeterminate result for the QFT because ofhigh negative control value (lines 204-205). In contrast, whereas IFN-g was undetectable in the negative control sample of the combined IGRA, results obtained in response to HBHA and to ESAT-6 were highly suggestive of active TB so that a PetScan was performed. This difference between the QFT and the combined IGRA has now been clarified in the text (lines 207-208).
  • Results from the combined HBHA/ESAT-6-IGRA are indeed slightly misleading for patient n°15. The IFN-g response to ESAT-6 was low albeit significant, and higher than the response to HBHA, which was unsignificant. Even though this profile is suggestive of active TB, it may also be associated to a poorly controlled latent TB and thus a higher risk of reactivation. Exclusion of other illnesses remains therefore important and in case of patient n°15, diagnosis of sarcoidosis was finally retained based on the high angiotensin convertase plasma levels. This point was clarified in the result section (lines 236-238) and the importance to still consider other blood results that might be helpful for the differential diagnosis was highlighted in the discussion (lines 321-327).

Line 96: Please specify concentrations of HBHA and ESAT-6 used.

Response: HBHA and ESAT-6 concentrations have now been added (line 101).

Line 124: According to Tables 1-3, age range was 18-79 years and not 18-66 years. Please correct.

Response: This mistake has now been corrected (line 133). We thank the reviewer for this comment.

Figure 1: Please specify in the foot note for group 2 that IGRA ESAT-6>HBHA result was indeterminate for 1 TBLA patient. Regarding the group 3, HBHA was negative for 2 patients (19 and 21) according to Table 4. Finally, it could be interesting to add statistical analysis to emphasize the added value of combined HBHA/ESAT-6 IGRA compared to IGRA tests alone. 

Response:

  • There was only one indeterminate result and this was for the QFT. In contrast, no indeterminate result was obtained for the combined HBHA/ESAT-6 IGRA. As the QFT results are not mentioned in Figure 1, the footnote was not modified.
  • We thank the reviewer for his/her comment about the number of patients from group 3 with a positive response to HBHA. There was indeed a mistake, as only 1/3 patient was positive : this mistake has been corrected in the new Figure 1.
  • Finally, as the QFT was only performed for a limited number of patients, we think that any statistical analysis comparing the results from the combined IGRA to those of the QFT would not be appropriate.

Lines 167-169: Please specify the group.

Response: As these lines are under the subtitle « 3.2. Diagnosis in group 1 : cervical and/ or axillar adenopathy », we feel it is not necessary to further specify the group.

Lines 172-174 and 266-267: The IGRA result ESAT-6<HBHA is indicated as characteristic of better control of Mtb infection, but according to the literature it also could be interpreted as the early stage of TB reactivation. Please clarify. 

Response: During latent TB, a positive IGRA result for both ESAT-6 and HBHA identifies subjects more at risk to reactivate Mtb infection than an IGRA result only positive for HBHA (Dirix V et al J Clin Microbiol 2022). However, within individuals with a positive IGRA for both antigens, a lower response to ESAT-6 than to HBHA characterizes better control of Mtb infection than higher response to ESAT-6 compared to HBHA (Mascart F et al, Expert Rev Vaccines 2015; Corbière V et al PLoS One 2012). Thus, an IGRA result ESAT-6>HBHA is suggestive of active TB or of poor control of the infection in case of latent TB (subjects with quiescent bacterial replication). In contrast, a limited IFN-g response to ESAT-6 is highly suggestive of active TB. We have now clarified this in the discussion (lines 316-327), and we have adapted the cited references in the results section.

Table 4: Instead of the column “clinical manifestation” it could be more informative to indicate the microbiological results to avoid redundancy with Tables 1-3. Please at least justify this choice.

Response: We thank the reviewer for this suggestion and Table 4 has been changed accordingly.

Lines 186-189: Please clarify why you consider HBHA/ESAT-6 IGRA result of the patient 12 as “highly suggestive of active TB”, despite the indeterminate result due to the elevated negative control. 

Response: As stated above, patient 12 had an indeterminate result only for the QFT. In contrast, for the combined HBHA/ESAT-6 IGRA, IFN-g was undetectable in the negative control so that the response to antigens could be interpreted. This has now been clarified (lines 207-208).

Discussion: As stated for the title, the results of the combined HBHA/ESAT-6 IGRA helped guide the diagnosis in some cases in this study, so the term "retrospective analysis" is not the most appropriate.

Response: Even though the study design was not that of a retrospective analysis, as some results were indeed used to guide the diagnosis, we present here the results from a retrospective analysis of the data once they were all available (clinical data, microbiology, histology, imagery, response to treatment). If we feel the term “retrospective” is indeed not appropriate in the title, we prefer to maintain it in the discussion as the final analysis of the results was retrospective.

Reviewer 2 Report

Dear authors, 

Congratulations for your research and for the employed workforce. Actually, extrapulmonary TB and specially TB adenopathy is an important issue. 

However, we must remember that if the patient presents with adenopathy, we could not consider him/her asymptomatic. Moreover, in high burden countries, many people get in contact t Mycobacterium tuberculosis without being aware of it. For this reason, the use of Blood-based interferon--release assays (IGRA) or tuberculin skin test (TST) as a diagnostic or triage method may not be helpful.  

Therefore, this application could be useful in quite specific scenarios after a large investigation. The use of the suggested strategy may lead healthcare workers to use IGRA or TST and misdiagnose TB. Missing diseases like cancer, fungal infections and even nontuberculous micobacteriosis. According to literature it may increase mortality rates once diseases like cancer, fungal infection or non-tuberculous mycobacteria are not investigated.  

Though, the data encountered is very interesting and may be developed in further research.

Yours faithfully,

Author Response

We must remember that if the patient presents with adenopathy, we could not consider him/her asymptomatic.

Response: Indeed, patients presenting with cervical adenopathy may not be considered as asymptomatic. In contrast, some patients with mediastinal adenopathy were strictly asymptomatic.  The adenopathy was discovered by PetScan performed in view of the IGRA results that were performed before starting anti-TNF-a treatment or by the health care worker department.

 Moreover, in high burden countries, many people get in contact with Mycobacterium tuberculosis without being aware of it. For this reason, the use of Blood-based interferon-g-release assays (IGRA) or tuberculin skin test (TST) as a diagnostic or triage method may not be helpful.  

Response: The commercialized blood-based interferon-g-release assays (IGRA), like the TST, do indeed not differentiate people who were in contact with Mtb from those who suffer from active TB. This is a major drawback of these tests. In contrast, the combined HBHA/ ESAT-6 IFN-g-release assay is the only IGRA providing a good, albeit not perfect, discrimination between latent and active TB, and allowing furthermore to stratify latently TB infected subjects in different subgroups more or less at risk to develop active TB. This is the reason why we propose here to use this combined HBHA/ESAT-6 IGRA as a triage test for the differential diagnosis of adenopathies.

 The use of the suggested strategy may lead healthcare workers to use IGRA or TST and misdiagnose TB. Missing diseases like cancer, fungal infections and even nontuberculous micobacteriosis. According to literature it may increase mortality rates once diseases like cancer, fungal infection or non-tuberculous mycobacteria are not investigated.  

Response: We do not suggest to exclusively base the differential diagnosis of adenopathy on results from the combined HBHA/ ESAT-6-IGRA. This test should be combined to other blood tests as for example measurement of angiotensin convertase concentrations in case of differential diagnosis with sarcoidosis. However, results from the combined IGRA are helpful in the differential diagnosis as :

  • Negative IGRA to both antigens, negative ESAT-6-IGRA with positive HBHA-IGRA, or positive IGRA to both antigens with a lower IFN-g response to ESAT-6 than to HBHA are not suggestive of active TB. The clinicians should consider other diagnosis.
  • In contrast, positive ESAT-6-IGRA with negative HBHA-IGRA is highly suggestive of active TB
  • And higher IFN-g response to ESAT-6 than to HBHA also highly suggest active TB. As this profile may also in some case be associated to quiescent bacterial replication in subjects with latent TB, other diagnosis should also be considered.

This is now more detailed in the discussion (lines 316-327).

Reviewer 3 Report

This is a well written and important article.  Retrospective analysis of a combined HBHA and ESAT-6-interferon-gamma release assay for diagnosis of TB lymphadenopathies.  This is important work that compares the IFN-gamma responses to two different stage-specific mycobacterial antigens.  The authors did a great deal of work in collecting the information from chart review for this analysis.This is a well written and important article.  Retrospective analysis of a combined HBHA and ESAT-6-interferon-gamma release assay for diagnosis of TB lymphadenopathies.  This is important work that compares the IFN-gamma responses to two different stage-specific mycobacterial antigens.  The authors did a great deal of work in collecting the information from chart review for this analysis.This is a well written and important article.  Retrospective analysis of a combined HBHA and ESAT-6-interferon-gamma release assay for diagnosis of TB lymphadenopathies.  This is important work that compares the IFN-gamma responses to two different stage-specific mycobacterial antigens.  The authors did a great deal of work in collecting the information from chart review for this analysis.This is a well written and important article.  Retrospective analysis of a combined HBHA and ESAT-6-interferon-gamma release assay for diagnosis of TB lymphadenopathies.  This is important work that compares the IFN-gamma responses to two different stage-specific mycobacterial antigens.  The authors did a great deal of work in collecting the information from chart review for this analysis.This is a well written and important article.  Retrospective analysis of a combined HBHA and ESAT-6-interferon-gamma release assay for diagnosis of TB lymphadenopathies.  This is important work that compares the IFN-gamma responses to two different stage-specific mycobacterial antigens.  The authors did a great deal of work in collecting the information from chart review for this analysis.This is a well written and important article.  Retrospective analysis of a combined HBHA and ESAT-6-interferon-gamma release assay for diagnosis of TB lymphadenopathies.  This is important work that compares the IFN-gamma responses to two different stage-specific mycobacterial antigens.  The authors did a great deal of work in collecting the information from chart review for this analysis.This is a well written and important article.  Retrospective analysis of a combined HBHA and ESAT-6-interferon-gamma release assay for diagnosis of TB lymphadenopathies.  This is important work that compares the IFN-gamma responses to two different stage-specific mycobacterial antigens.  The authors did a great deal of work in collecting the information from chart review for this analysis.This is a well written and important article.  Retrospective analysis of a combined HBHA and ESAT-6-interferon-gamma release assay for diagnosis of TB lymphadenopathies.  This is important work that compares the IFN-gamma responses to two different stage-specific mycobacterial antigens.  The authors did a great deal of work in collecting the information from chart review for this analysis. 

This is a retrospective evaluation of a blood-cell interferon-g release 29 assay (IGRA) with two different stage-specific mycobacterial antigens for the differential diagnosis of ADP suspected of mycobacterial origin.  This can be a major improvement since active/subclinical TB can be differentiated from latent TB by using these antigens.  There needs to be a great deal of further studies before it can be routinely used.  The time savings over waiting for cultures and the greater sensitivity over culture makes this important work. 

This study included a small sample size, N = 21, suspected to present TBLA without pulmonary involvement.  The authors broke the sample into three groups: N =8 cervical/axillary ADP, N = 10 mediastinal ADP, and N = 3 disseminated ADP. 

The authors were able to provide results for 15 patients.  The results of the combined HBHA and ESAT-6-interferon-gamma release assay were compared to culture.  This is a little problematic since culture is not truly a gold standard but this is all we have to compare results to.  Three of 6 had IGRA profile suggestive of active TB. Results are not noted for patients with alternative diagnosis determined.

Conclusion – HBH/ESAT-6 IGRA as a triage test has potential

There is some gaps in data, which are understandable, but the authors are cautious in their conclusions.

The authors address the limitation of the study. 

Recommendations from the reviewer:

·         In the methods section, the authors should add more details about the HBHA assay “culture medium” because reference 10 cites another reference and I could not find specifics in that reference

·         In tables 1-3 under Risk factor, if it is blank, was the information not available?  Please make this clear to the reader. 

·         In figure 1, please clarify what “doubtful” histology results mean.

·         Minor: Line 348 #10 is repeated for this reference.

I recommend acceptance after minor edits.This is a retrospective evaluation of a blood-cell interferon-g release 29 assay (IGRA) with two different stage-specific mycobacterial antigens for the differential diagnosis of ADP suspected of mycobacterial origin.  This can be a major improvement since active/subclinical TB can be differentiated from latent TB by using these antigens.  There needs to be a great deal of further studies before it can be routinely used.  The time savings over waiting for cultures and the greater sensitivity over culture makes this important work. 

This study included a small sample size, N = 21, suspected to present TBLA without pulmonary involvement.  The authors broke the sample into three groups: N =8 cervical/axillary ADP, N = 10 mediastinal ADP, and N = 3 disseminated ADP. 

The authors were able to provide results for 15 patients.  The results of the combined HBHA and ESAT-6-interferon-gamma release assay were compared to culture.  This is a little problematic since culture is not truly a gold standard but this is all we have to compare results to.  Three of 6 had IGRA profile suggestive of active TB. Results are not noted for patients with alternative diagnosis determined.

Conclusion – HBH/ESAT-6 IGRA as a triage test has potential

There is some gaps in data, which are understandable, but the authors are cautious in their conclusions.

The authors address the limitation of the study. 

Recommendations from the reviewer:

·         In the methods section, the authors should add more details about the HBHA assay “culture medium” because reference 10 cites another reference and I could not find specifics in that reference

·         In tables 1-3 under Risk factor, if it is blank, was the information not available?  Please make this clear to the reader. 

·         In figure 1, please clarify what “doubtful” histology results mean.

·         Minor: Line 348 #10 is repeated for this reference.

I recommend acceptance after minor edits.This is a retrospective evaluation of a blood-cell interferon-g release 29 assay (IGRA) with two different stage-specific mycobacterial antigens for the differential diagnosis of ADP suspected of mycobacterial origin.  This can be a major improvement since active/subclinical TB can be differentiated from latent TB by using these antigens.  There needs to be a great deal of further studies before it can be routinely used.  The time savings over waiting for cultures and the greater sensitivity over culture makes this important work. 

This study included a small sample size, N = 21, suspected to present TBLA without pulmonary involvement.  The authors broke the sample into three groups: N =8 cervical/axillary ADP, N = 10 mediastinal ADP, and N = 3 disseminated ADP. 

The authors were able to provide results for 15 patients.  The results of the combined HBHA and ESAT-6-interferon-gamma release assay were compared to culture.  This is a little problematic since culture is not truly a gold standard but this is all we have to compare results to.  Three of 6 had IGRA profile suggestive of active TB. Results are not noted for patients with alternative diagnosis determined.

Conclusion – HBH/ESAT-6 IGRA as a triage test has potential

There is some gaps in data, which are understandable, but the authors are cautious in their conclusions.

The authors address the limitation of the study. 

Recommendations from the reviewer:

·         In the methods section, the authors should add more details about the HBHA assay “culture medium” because reference 10 cites another reference and I could not find specifics in that reference

·         In tables 1-3 under Risk factor, if it is blank, was the information not available?  Please make this clear to the reader. 

·         In figure 1, please clarify what “doubtful” histology results mean.

·         Minor: Line 348 #10 is repeated for this reference.

I recommend acceptance after minor edits.This is a retrospective evaluation of a blood-cell interferon-g release 29 assay (IGRA) with two different stage-specific mycobacterial antigens for the differential diagnosis of ADP suspected of mycobacterial origin.  This can be a major improvement since active/subclinical TB can be differentiated from latent TB by using these antigens.  There needs to be a great deal of further studies before it can be routinely used.  The time savings over waiting for cultures and the greater sensitivity over culture makes this important work. 

This study included a small sample size, N = 21, suspected to present TBLA without pulmonary involvement.  The authors broke the sample into three groups: N =8 cervical/axillary ADP, N = 10 mediastinal ADP, and N = 3 disseminated ADP. 

The authors were able to provide results for 15 patients.  The results of the combined HBHA and ESAT-6-interferon-gamma release assay were compared to culture.  This is a little problematic since culture is not truly a gold standard but this is all we have to compare results to.  Three of 6 had IGRA profile suggestive of active TB. Results are not noted for patients with alternative diagnosis determined.

Conclusion – HBH/ESAT-6 IGRA as a triage test has potential

There is some gaps in data, which are understandable, but the authors are cautious in their conclusions.

The authors address the limitation of the study. 

Recommendations from the reviewer:

·         In the methods section, the authors should add more details about the HBHA assay “culture medium” because reference 10 cites another reference and I could not find specifics in that reference

·         In tables 1-3 under Risk factor, if it is blank, was the information not available?  Please make this clear to the reader. 

·         In figure 1, please clarify what “doubtful” histology results mean.

·         Minor: Line 348 #10 is repeated for this reference.

I recommend acceptance after minor edits.This is a retrospective evaluation of a blood-cell interferon-g release 29 assay (IGRA) with two different stage-specific mycobacterial antigens for the differential diagnosis of ADP suspected of mycobacterial origin.  This can be a major improvement since active/subclinical TB can be differentiated from latent TB by using these antigens.  There needs to be a great deal of further studies before it can be routinely used.  The time savings over waiting for cultures and the greater sensitivity over culture makes this important work. 

This study included a small sample size, N = 21, suspected to present TBLA without pulmonary involvement.  The authors broke the sample into three groups: N =8 cervical/axillary ADP, N = 10 mediastinal ADP, and N = 3 disseminated ADP. 

The authors were able to provide results for 15 patients.  The results of the combined HBHA and ESAT-6-interferon-gamma release assay were compared to culture.  This is a little problematic since culture is not truly a gold standard but this is all we have to compare results to.  Three of 6 had IGRA profile suggestive of active TB. Results are not noted for patients with alternative diagnosis determined.

Conclusion – HBH/ESAT-6 IGRA as a triage test has potential

There is some gaps in data, which are understandable, but the authors are cautious in their conclusions.

The authors address the limitation of the study. 

Recommendations from the reviewer:

·         In the methods section, the authors should add more details about the HBHA assay “culture medium” because reference 10 cites another reference and I could not find specifics in that reference

·         In tables 1-3 under Risk factor, if it is blank, was the information not available?  Please make this clear to the reader. 

·         In figure 1, please clarify what “doubtful” histology results mean.

·         Minor: Line 348 #10 is repeated for this reference.

I recommend acceptance after minor edits.This is a retrospective evaluation of a blood-cell interferon-g release 29 assay (IGRA) with two different stage-specific mycobacterial antigens for the differential diagnosis of ADP suspected of mycobacterial origin.  This can be a major improvement since active/subclinical TB can be differentiated from latent TB by using these antigens.  There needs to be a great deal of further studies before it can be routinely used.  The time savings over waiting for cultures and the greater sensitivity over culture makes this important work. 

This study included a small sample size, N = 21, suspected to present TBLA without pulmonary involvement.  The authors broke the sample into three groups: N =8 cervical/axillary ADP, N = 10 mediastinal ADP, and N = 3 disseminated ADP. 

The authors were able to provide results for 15 patients.  The results of the combined HBHA and ESAT-6-interferon-gamma release assay were compared to culture.  This is a little problematic since culture is not truly a gold standard but this is all we have to compare results to.  Three of 6 had IGRA profile suggestive of active TB. Results are not noted for patients with alternative diagnosis determined.

Conclusion – HBH/ESAT-6 IGRA as a triage test has potential

There is some gaps in data, which are understandable, but the authors are cautious in their conclusions.

The authors address the limitation of the study. 

Recommendations from the reviewer:

·         In the methods section, the authors should add more details about the HBHA assay “culture medium” because reference 10 cites another reference and I could not find specifics in that reference

·         In tables 1-3 under Risk factor, if it is blank, was the information not available?  Please make this clear to the reader. 

·         In figure 1, please clarify what “doubtful” histology results mean.

·         Minor: Line 348 #10 is repeated for this reference.

I recommend acceptance after minor edits.This is a retrospective evaluation of a blood-cell interferon-g release 29 assay (IGRA) with two different stage-specific mycobacterial antigens for the differential diagnosis of ADP suspected of mycobacterial origin.  This can be a major improvement since active/subclinical TB can be differentiated from latent TB by using these antigens.  There needs to be a great deal of further studies before it can be routinely used.  The time savings over waiting for cultures and the greater sensitivity over culture makes this important work. 

This study included a small sample size, N = 21, suspected to present TBLA without pulmonary involvement.  The authors broke the sample into three groups: N =8 cervical/axillary ADP, N = 10 mediastinal ADP, and N = 3 disseminated ADP. 

The authors were able to provide results for 15 patients.  The results of the combined HBHA and ESAT-6-interferon-gamma release assay were compared to culture.  This is a little problematic since culture is not truly a gold standard but this is all we have to compare results to.  Three of 6 had IGRA profile suggestive of active TB. Results are not noted for patients with alternative diagnosis determined.

Conclusion – HBH/ESAT-6 IGRA as a triage test has potential

There is some gaps in data, which are understandable, but the authors are cautious in their conclusions.

The authors address the limitation of the study. 

Recommendations from the reviewer:

·         In the methods section, the authors should add more details about the HBHA assay “culture medium” because reference 10 cites another reference and I could not find specifics in that reference

·         In tables 1-3 under Risk factor, if it is blank, was the information not available?  Please make this clear to the reader. 

·         In figure 1, please clarify what “doubtful” histology results mean.

·         Minor: Line 348 #10 is repeated for this reference.

I recommend acceptance after minor edits.This is a retrospective evaluation of a blood-cell interferon-g release 29 assay (IGRA) with two different stage-specific mycobacterial antigens for the differential diagnosis of ADP suspected of mycobacterial origin.  This can be a major improvement since active/subclinical TB can be differentiated from latent TB by using these antigens.  There needs to be a great deal of further studies before it can be routinely used.  The time savings over waiting for cultures and the greater sensitivity over culture makes this important work. 

This study included a small sample size, N = 21, suspected to present TBLA without pulmonary involvement.  The authors broke the sample into three groups: N =8 cervical/axillary ADP, N = 10 mediastinal ADP, and N = 3 disseminated ADP. 

The authors were able to provide results for 15 patients.  The results of the combined HBHA and ESAT-6-interferon-gamma release assay were compared to culture.  This is a little problematic since culture is not truly a gold standard but this is all we have to compare results to.  Three of 6 had IGRA profile suggestive of active TB. Results are not noted for patients with alternative diagnosis determined.

Conclusion – HBH/ESAT-6 IGRA as a triage test has potential

There is some gaps in data, which are understandable, but the authors are cautious in their conclusions.

The authors address the limitation of the study. 

Recommendations from the reviewer:

·         In the methods section, the authors should add more details about the HBHA assay “culture medium” because reference 10 cites another reference and I could not find specifics in that reference

·         In tables 1-3 under Risk factor, if it is blank, was the information not available?  Please make this clear to the reader. 

·         In figure 1, please clarify what “doubtful” histology results mean.

·         Minor: Line 348 #10 is repeated for this reference.

I recommend acceptance after minor edits.This is a retrospective evaluation of a blood-cell interferon-g release 29 assay (IGRA) with two different stage-specific mycobacterial antigens for the differential diagnosis of ADP suspected of mycobacterial origin.  This can be a major improvement since active/subclinical TB can be differentiated from latent TB by using these antigens.  There needs to be a great deal of further studies before it can be routinely used.  The time savings over waiting for cultures and the greater sensitivity over culture makes this important work. 

This study included a small sample size, N = 21, suspected to present TBLA without pulmonary involvement.  The authors broke the sample into three groups: N =8 cervical/axillary ADP, N = 10 mediastinal ADP, and N = 3 disseminated ADP. 

The authors were able to provide results for 15 patients.  The results of the combined HBHA and ESAT-6-interferon-gamma release assay were compared to culture.  This is a little problematic since culture is not truly a gold standard but this is all we have to compare results to.  Three of 6 had IGRA profile suggestive of active TB. Results are not noted for patients with alternative diagnosis determined.

Conclusion – HBH/ESAT-6 IGRA as a triage test has potential

There is some gaps in data, which are understandable, but the authors are cautious in their conclusions.

The authors address the limitation of the study. 

Recommendations from the reviewer:

·         In the methods section, the authors should add more details about the HBHA assay “culture medium” because reference 10 cites another reference and I could not find specifics in that reference

·         In tables 1-3 under Risk factor, if it is blank, was the information not available?  Please make this clear to the reader. 

·         In figure 1, please clarify what “doubtful” histology results mean.

·         Minor: Line 348 #10 is repeated for this reference.

I recommend acceptance after minor edits.This is a retrospective evaluation of a blood-cell interferon-g release 29 assay (IGRA) with two different stage-specific mycobacterial antigens for the differential diagnosis of ADP suspected of mycobacterial origin.  This can be a major improvement since active/subclinical TB can be differentiated from latent TB by using these antigens.  There needs to be a great deal of further studies before it can be routinely used.  The time savings over waiting for cultures and the greater sensitivity over culture makes this important work. 

This study included a small sample size, N = 21, suspected to present TBLA without pulmonary involvement.  The authors broke the sample into three groups: N =8 cervical/axillary ADP, N = 10 mediastinal ADP, and N = 3 disseminated ADP. 

The authors were able to provide results for 15 patients.  The results of the combined HBHA and ESAT-6-interferon-gamma release assay were compared to culture.  This is a little problematic since culture is not truly a gold standard but this is all we have to compare results to.  Three of 6 had IGRA profile suggestive of active TB. Results are not noted for patients with alternative diagnosis determined.

Conclusion – HBH/ESAT-6 IGRA as a triage test has potential

There is some gaps in data, which are understandable, but the authors are cautious in their conclusions.

The authors address the limitation of the study. 

Recommendations from the reviewer:

·         In the methods section, the authors should add more details about the HBHA assay “culture medium” because reference 10 cites another reference and I could not find specifics in that reference

·         In tables 1-3 under Risk factor, if it is blank, was the information not available?  Please make this clear to the reader. 

·         In figure 1, please clarify what “doubtful” histology results mean.

·         Minor: Line 348 #10 is repeated for this reference.

I recommend acceptance after minor edits.This is a retrospective evaluation of a blood-cell interferon-g release 29 assay (IGRA) with two different stage-specific mycobacterial antigens for the differential diagnosis of ADP suspected of mycobacterial origin.  This can be a major improvement since active/subclinical TB can be differentiated from latent TB by using these antigens.  There needs to be a great deal of further studies before it can be routinely used.  The time savings over waiting for cultures and the greater sensitivity over culture makes this important work. 

This study included a small sample size, N = 21, suspected to present TBLA without pulmonary involvement.  The authors broke the sample into three groups: N =8 cervical/axillary ADP, N = 10 mediastinal ADP, and N = 3 disseminated ADP. 

The authors were able to provide results for 15 patients.  The results of the combined HBHA and ESAT-6-interferon-gamma release assay were compared to culture.  This is a little problematic since culture is not truly a gold standard but this is all we have to compare results to.  Three of 6 had IGRA profile suggestive of active TB. Results are not noted for patients with alternative diagnosis determined.

Conclusion – HBH/ESAT-6 IGRA as a triage test has potential

There is some gaps in data, which are understandable, but the authors are cautious in their conclusions.

The authors address the limitation of the study. 

Recommendations from the reviewer:

·         In the methods section, the authors should add more details about the HBHA assay “culture medium” because reference 10 cites another reference and I could not find specifics in that reference

·         In tables 1-3 under Risk factor, if it is blank, was the information not available?  Please make this clear to the reader. 

·         In figure 1, please clarify what “doubtful” histology results mean.

·         Minor: Line 348 #10 is repeated for this reference.

I recommend acceptance after minor edits.This is a retrospective evaluation of a blood-cell interferon-g release 29 assay (IGRA) with two different stage-specific mycobacterial antigens for the differential diagnosis of ADP suspected of mycobacterial origin.  This can be a major improvement since active/subclinical TB can be differentiated from latent TB by using these antigens.  There needs to be a great deal of further studies before it can be routinely used.  The time savings over waiting for cultures and the greater sensitivity over culture makes this important work. 

This study included a small sample size, N = 21, suspected to present TBLA without pulmonary involvement.  The authors broke the sample into three groups: N =8 cervical/axillary ADP, N = 10 mediastinal ADP, and N = 3 disseminated ADP. 

The authors were able to provide results for 15 patients.  The results of the combined HBHA and ESAT-6-interferon-gamma release assay were compared to culture.  This is a little problematic since culture is not truly a gold standard but this is all we have to compare results to.  Three of 6 had IGRA profile suggestive of active TB. Results are not noted for patients with alternative diagnosis determined.

Conclusion – HBH/ESAT-6 IGRA as a triage test has potential

There is some gaps in data, which are understandable, but the authors are cautious in their conclusions.

The authors address the limitation of the study. 

Recommendations from the reviewer:

·         In the methods section, the authors should add more details about the HBHA assay “culture medium” because reference 10 cites another reference and I could not find specifics in that reference

·         In tables 1-3 under Risk factor, if it is blank, was the information not available?  Please make this clear to the reader. 

·         In figure 1, please clarify what “doubtful” histology results mean.

·         Minor: Line 348 #10 is repeated for this reference.

I recommend acceptance after minor edits.This is a retrospective evaluation of a blood-cell interferon-g release 29 assay (IGRA) with two different stage-specific mycobacterial antigens for the differential diagnosis of ADP suspected of mycobacterial origin.  This can be a major improvement since active/subclinical TB can be differentiated from latent TB by using these antigens.  There needs to be a great deal of further studies before it can be routinely used.  The time savings over waiting for cultures and the greater sensitivity over culture makes this important work. 

This study included a small sample size, N = 21, suspected to present TBLA without pulmonary involvement.  The authors broke the sample into three groups: N =8 cervical/axillary ADP, N = 10 mediastinal ADP, and N = 3 disseminated ADP. 

The authors were able to provide results for 15 patients.  The results of the combined HBHA and ESAT-6-interferon-gamma release assay were compared to culture.  This is a little problematic since culture is not truly a gold standard but this is all we have to compare results to.  Three of 6 had IGRA profile suggestive of active TB. Results are not noted for patients with alternative diagnosis determined.

Conclusion – HBH/ESAT-6 IGRA as a triage test has potential

There is some gaps in data, which are understandable, but the authors are cautious in their conclusions.

The authors address the limitation of the study. 

Recommendations from the reviewer:

·         In the methods section, the authors should add more details about the HBHA assay “culture medium” because reference 10 cites another reference and I could not find specifics in that reference

·         In tables 1-3 under Risk factor, if it is blank, was the information not available?  Please make this clear to the reader. 

·         In figure 1, please clarify what “doubtful” histology results mean.

·         Minor: Line 348 #10 is repeated for this reference.

I recommend acceptance after minor edits.This is a retrospective evaluation of a blood-cell interferon-g release 29 assay (IGRA) with two different stage-specific mycobacterial antigens for the differential diagnosis of ADP suspected of mycobacterial origin.  This can be a major improvement since active/subclinical TB can be differentiated from latent TB by using these antigens.  There needs to be a great deal of further studies before it can be routinely used.  The time savings over waiting for cultures and the greater sensitivity over culture makes this important work. 

This study included a small sample size, N = 21, suspected to present TBLA without pulmonary involvement.  The authors broke the sample into three groups: N =8 cervical/axillary ADP, N = 10 mediastinal ADP, and N = 3 disseminated ADP. 

The authors were able to provide results for 15 patients.  The results of the combined HBHA and ESAT-6-interferon-gamma release assay were compared to culture.  This is a little problematic since culture is not truly a gold standard but this is all we have to compare results to.  Three of 6 had IGRA profile suggestive of active TB. Results are not noted for patients with alternative diagnosis determined.

Conclusion – HBH/ESAT-6 IGRA as a triage test has potential

There is some gaps in data, which are understandable, but the authors are cautious in their conclusions.

The authors address the limitation of the study. 

Recommendations from the reviewer:

·         In the methods section, the authors should add more details about the HBHA assay “culture medium” because reference 10 cites another reference and I could not find specifics in that reference

·         In tables 1-3 under Risk factor, if it is blank, was the information not available?  Please make this clear to the reader. 

·         In figure 1, please clarify what “doubtful” histology results mean.

·         Minor: Line 348 #10 is repeated for this reference.

I recommend acceptance after minor edits.This is a retrospective evaluation of a blood-cell interferon-g release 29 assay (IGRA) with two different stage-specific mycobacterial antigens for the differential diagnosis of ADP suspected of mycobacterial origin.  This can be a major improvement since active/subclinical TB can be differentiated from latent TB by using these antigens.  There needs to be a great deal of further studies before it can be routinely used.  The time savings over waiting for cultures and the greater sensitivity over culture makes this important work. 

This study included a small sample size, N = 21, suspected to present TBLA without pulmonary involvement.  The authors broke the sample into three groups: N =8 cervical/axillary ADP, N = 10 mediastinal ADP, and N = 3 disseminated ADP. 

The authors were able to provide results for 15 patients.  The results of the combined HBHA and ESAT-6-interferon-gamma release assay were compared to culture.  This is a little problematic since culture is not truly a gold standard but this is all we have to compare results to.  Three of 6 had IGRA profile suggestive of active TB. Results are not noted for patients with alternative diagnosis determined.

Conclusion – HBH/ESAT-6 IGRA as a triage test has potential

There is some gaps in data, which are understandable, but the authors are cautious in their conclusions.

The authors address the limitation of the study. 

Recommendations from the reviewer:

·         In the methods section, the authors should add more details about the HBHA assay “culture medium” because reference 10 cites another reference and I could not find specifics in that reference

·         In tables 1-3 under Risk factor, if it is blank, was the information not available?  Please make this clear to the reader. 

·         In figure 1, please clarify what “doubtful” histology results mean.

·         Minor: Line 348 #10 is repeated for this reference.

I recommend acceptance after minor edits.This is a retrospective evaluation of a blood-cell interferon-g release 29 assay (IGRA) with two different stage-specific mycobacterial antigens for the differential diagnosis of ADP suspected of mycobacterial origin.  This can be a major improvement since active/subclinical TB can be differentiated from latent TB by using these antigens.  There needs to be a great deal of further studies before it can be routinely used.  The time savings over waiting for cultures and the greater sensitivity over culture makes this important work. 

This study included a small sample size, N = 21, suspected to present TBLA without pulmonary involvement.  The authors broke the sample into three groups: N =8 cervical/axillary ADP, N = 10 mediastinal ADP, and N = 3 disseminated ADP. 

The authors were able to provide results for 15 patients.  The results of the combined HBHA and ESAT-6-interferon-gamma release assay were compared to culture.  This is a little problematic since culture is not truly a gold standard but this is all we have to compare results to.  Three of 6 had IGRA profile suggestive of active TB. Results are not noted for patients with alternative diagnosis determined.

Conclusion – HBH/ESAT-6 IGRA as a triage test has potential

There is some gaps in data, which are understandable, but the authors are cautious in their conclusions.

The authors address the limitation of the study. 

Recommendations from the reviewer:

·         In the methods section, the authors should add more details about the HBHA assay “culture medium” because reference 10 cites another reference and I could not find specifics in that reference

·         In tables 1-3 under Risk factor, if it is blank, was the information not available?  Please make this clear to the reader. 

·         In figure 1, please clarify what “doubtful” histology results mean.

·         Minor: Line 348 #10 is repeated for this reference.

I recommend acceptance after minor edits.This is a retrospective evaluation of a blood-cell interferon-g release 29 assay (IGRA) with two different stage-specific mycobacterial antigens for the differential diagnosis of ADP suspected of mycobacterial origin.  This can be a major improvement since active/subclinical TB can be differentiated from latent TB by using these antigens.  There needs to be a great deal of further studies before it can be routinely used.  The time savings over waiting for cultures and the greater sensitivity over culture makes this important work. 

This study included a small sample size, N = 21, suspected to present TBLA without pulmonary involvement.  The authors broke the sample into three groups: N =8 cervical/axillary ADP, N = 10 mediastinal ADP, and N = 3 disseminated ADP. 

The authors were able to provide results for 15 patients.  The results of the combined HBHA and ESAT-6-interferon-gamma release assay were compared to culture.  This is a little problematic since culture is not truly a gold standard but this is all we have to compare results to.  Three of 6 had IGRA profile suggestive of active TB. Results are not noted for patients with alternative diagnosis determined.

Conclusion – HBH/ESAT-6 IGRA as a triage test has potential

There is some gaps in data, which are understandable, but the authors are cautious in their conclusions.

The authors address the limitation of the study. 

Recommendations from the reviewer:

·         In the methods section, the authors should add more details about the HBHA assay “culture medium” because reference 10 cites another reference and I could not find specifics in that reference

·         In tables 1-3 under Risk factor, if it is blank, was the information not available?  Please make this clear to the reader. 

·         In figure 1, please clarify what “doubtful” histology results mean.

·         Minor: Line 348 #10 is repeated for this reference.

I recommend acceptance after minor edits.This is a retrospective evaluation of a blood-cell interferon-g release 29 assay (IGRA) with two different stage-specific mycobacterial antigens for the differential diagnosis of ADP suspected of mycobacterial origin.  This can be a major improvement since active/subclinical TB can be differentiated from latent TB by using these antigens.  There needs to be a great deal of further studies before it can be routinely used.  The time savings over waiting for cultures and the greater sensitivity over culture makes this important work. 

This study included a small sample size, N = 21, suspected to present TBLA without pulmonary involvement.  The authors broke the sample into three groups: N =8 cervical/axillary ADP, N = 10 mediastinal ADP, and N = 3 disseminated ADP. 

The authors were able to provide results for 15 patients.  The results of the combined HBHA and ESAT-6-interferon-gamma release assay were compared to culture.  This is a little problematic since culture is not truly a gold standard but this is all we have to compare results to.  Three of 6 had IGRA profile suggestive of active TB. Results are not noted for patients with alternative diagnosis determined.

Conclusion – HBH/ESAT-6 IGRA as a triage test has potential

There is some gaps in data, which are understandable, but the authors are cautious in their conclusions.

The authors address the limitation of the study. 

Recommendations from the reviewer:

·         In the methods section, the authors should add more details about the HBHA assay “culture medium” because reference 10 cites another reference and I could not find specifics in that reference

·         In tables 1-3 under Risk factor, if it is blank, was the information not available?  Please make this clear to the reader. 

·         In figure 1, please clarify what “doubtful” histology results mean.

·         Minor: Line 348 #10 is repeated for this reference.

I recommend acceptance after minor edits.This is a retrospective evaluation of a blood-cell interferon-g release 29 assay (IGRA) with two different stage-specific mycobacterial antigens for the differential diagnosis of ADP suspected of mycobacterial origin.  This can be a major improvement since active/subclinical TB can be differentiated from latent TB by using these antigens.  There needs to be a great deal of further studies before it can be routinely used.  The time savings over waiting for cultures and the greater sensitivity over culture makes this important work. 

This study included a small sample size, N = 21, suspected to present TBLA without pulmonary involvement.  The authors broke the sample into three groups: N =8 cervical/axillary ADP, N = 10 mediastinal ADP, and N = 3 disseminated ADP. 

The authors were able to provide results for 15 patients.  The results of the combined HBHA and ESAT-6-interferon-gamma release assay were compared to culture.  This is a little problematic since culture is not truly a gold standard but this is all we have to compare results to.  Three of 6 had IGRA profile suggestive of active TB. Results are not noted for patients with alternative diagnosis determined.

Conclusion – HBH/ESAT-6 IGRA as a triage test has potential

There is some gaps in data, which are understandable, but the authors are cautious in their conclusions.

The authors address the limitation of the study. 

Recommendations from the reviewer:

·         In the methods section, the authors should add more details about the HBHA assay “culture medium” because reference 10 cites another reference and I could not find specifics in that reference

·         In tables 1-3 under Risk factor, if it is blank, was the information not available?  Please make this clear to the reader. 

·         In figure 1, please clarify what “doubtful” histology results mean.

·         Minor: Line 348 #10 is repeated for this reference.

I recommend acceptance after minor edits.This is a retrospective evaluation of a blood-cell interferon-g release 29 assay (IGRA) with two different stage-specific mycobacterial antigens for the differential diagnosis of ADP suspected of mycobacterial origin.  This can be a major improvement since active/subclinical TB can be differentiated from latent TB by using these antigens.  There needs to be a great deal of further studies before it can be routinely used.  The time savings over waiting for cultures and the greater sensitivity over culture makes this important work. 

This study included a small sample size, N = 21, suspected to present TBLA without pulmonary involvement.  The authors broke the sample into three groups: N =8 cervical/axillary ADP, N = 10 mediastinal ADP, and N = 3 disseminated ADP. 

The authors were able to provide results for 15 patients.  The results of the combined HBHA and ESAT-6-interferon-gamma release assay were compared to culture.  This is a little problematic since culture is not truly a gold standard but this is all we have to compare results to.  Three of 6 had IGRA profile suggestive of active TB. Results are not noted for patients with alternative diagnosis determined.

Conclusion – HBH/ESAT-6 IGRA as a triage test has potential

There is some gaps in data, which are understandable, but the authors are cautious in their conclusions.

The authors address the limitation of the study. 

Recommendations from the reviewer:

·         In the methods section, the authors should add more details about the HBHA assay “culture medium” because reference 10 cites another reference and I could not find specifics in that reference

·         In tables 1-3 under Risk factor, if it is blank, was the information not available?  Please make this clear to the reader. 

·         In figure 1, please clarify what “doubtful” histology results mean.

·         Minor: Line 348 #10 is repeated for this reference.

I recommend acceptance after minor edits.This is a retrospective evaluation of a blood-cell interferon-g release 29 assay (IGRA) with two different stage-specific mycobacterial antigens for the differential diagnosis of ADP suspected of mycobacterial origin.  This can be a major improvement since active/subclinical TB can be differentiated from latent TB by using these antigens.  There needs to be a great deal of further studies before it can be routinely used.  The time savings over waiting for cultures and the greater sensitivity over culture makes this important work. 

This study included a small sample size, N = 21, suspected to present TBLA without pulmonary involvement.  The authors broke the sample into three groups: N =8 cervical/axillary ADP, N = 10 mediastinal ADP, and N = 3 disseminated ADP. 

The authors were able to provide results for 15 patients.  The results of the combined HBHA and ESAT-6-interferon-gamma release assay were compared to culture.  This is a little problematic since culture is not truly a gold standard but this is all we have to compare results to.  Three of 6 had IGRA profile suggestive of active TB. Results are not noted for patients with alternative diagnosis determined.

Conclusion – HBH/ESAT-6 IGRA as a triage test has potential

There is some gaps in data, which are understandable, but the authors are cautious in their conclusions.

The authors address the limitation of the study. 

Recommendations from the reviewer:

·         In the methods section, the authors should add more details about the HBHA assay “culture medium” because reference 10 cites another reference and I could not find specifics in that reference

·         In tables 1-3 under Risk factor, if it is blank, was the information not available?  Please make this clear to the reader. 

·         In figure 1, please clarify what “doubtful” histology results mean.

·         Minor: Line 348 #10 is repeated for this reference.

I recommend acceptance after minor edits.This is a retrospective evaluation of a blood-cell interferon-g release 29 assay (IGRA) with two different stage-specific mycobacterial antigens for the differential diagnosis of ADP suspected of mycobacterial origin.  This can be a major improvement since active/subclinical TB can be differentiated from latent TB by using these antigens.  There needs to be a great deal of further studies before it can be routinely used.  The time savings over waiting for cultures and the greater sensitivity over culture makes this important work. 

This study included a small sample size, N = 21, suspected to present TBLA without pulmonary involvement.  The authors broke the sample into three groups: N =8 cervical/axillary ADP, N = 10 mediastinal ADP, and N = 3 disseminated ADP. 

The authors were able to provide results for 15 patients.  The results of the combined HBHA and ESAT-6-interferon-gamma release assay were compared to culture.  This is a little problematic since culture is not truly a gold standard but this is all we have to compare results to.  Three of 6 had IGRA profile suggestive of active TB. Results are not noted for patients with alternative diagnosis determined.

Conclusion – HBH/ESAT-6 IGRA as a triage test has potential

There is some gaps in data, which are understandable, but the authors are cautious in their conclusions.

The authors address the limitation of the study. 

Recommendations from the reviewer:

·         In the methods section, the authors should add more details about the HBHA assay “culture medium” because reference 10 cites another reference and I could not find specifics in that reference

·         In tables 1-3 under Risk factor, if it is blank, was the information not available?  Please make this clear to the reader. 

·         In figure 1, please clarify what “doubtful” histology results mean.

·         Minor: Line 348 #10 is repeated for this reference.

I recommend acceptance after minor edits.This is a retrospective evaluation of a blood-cell interferon-g release 29 assay (IGRA) with two different stage-specific mycobacterial antigens for the differential diagnosis of ADP suspected of mycobacterial origin.  This can be a major improvement since active/subclinical TB can be differentiated from latent TB by using these antigens.  There needs to be a great deal of further studies before it can be routinely used.  The time savings over waiting for cultures and the greater sensitivity over culture makes this important work. 

This study included a small sample size, N = 21, suspected to present TBLA without pulmonary involvement.  The authors broke the sample into three groups: N =8 cervical/axillary ADP, N = 10 mediastinal ADP, and N = 3 disseminated ADP. 

The authors were able to provide results for 15 patients.  The results of the combined HBHA and ESAT-6-interferon-gamma release assay were compared to culture.  This is a little problematic since culture is not truly a gold standard but this is all we have to compare results to.  Three of 6 had IGRA profile suggestive of active TB. Results are not noted for patients with alternative diagnosis determined.

Conclusion – HBH/ESAT-6 IGRA as a triage test has potential

There is some gaps in data, which are understandable, but the authors are cautious in their conclusions.

The authors address the limitation of the study. 

Recommendations from the reviewer:

·         In the methods section, the authors should add more details about the HBHA assay “culture medium” because reference 10 cites another reference and I could not find specifics in that reference

·         In tables 1-3 under Risk factor, if it is blank, was the information not available?  Please make this clear to the reader. 

·         In figure 1, please clarify what “doubtful” histology results mean.

·         Minor: Line 348 #10 is repeated for this reference.

I recommend acceptance after minor edits.This is a retrospective evaluation of a blood-cell interferon-g release 29 assay (IGRA) with two different stage-specific mycobacterial antigens for the differential diagnosis of ADP suspected of mycobacterial origin.  This can be a major improvement since active/subclinical TB can be differentiated from latent TB by using these antigens.  There needs to be a great deal of further studies before it can be routinely used.  The time savings over waiting for cultures and the greater sensitivity over culture makes this important work. 

This study included a small sample size, N = 21, suspected to present TBLA without pulmonary involvement.  The authors broke the sample into three groups: N =8 cervical/axillary ADP, N = 10 mediastinal ADP, and N = 3 disseminated ADP. 

The authors were able to provide results for 15 patients.  The results of the combined HBHA and ESAT-6-interferon-gamma release assay were compared to culture.  This is a little problematic since culture is not truly a gold standard but this is all we have to compare results to.  Three of 6 had IGRA profile suggestive of active TB. Results are not noted for patients with alternative diagnosis determined.

Conclusion – HBH/ESAT-6 IGRA as a triage test has potential

There is some gaps in data, which are understandable, but the authors are cautious in their conclusions.

The authors address the limitation of the study. 

Recommendations from the reviewer:

·         In the methods section, the authors should add more details about the HBHA assay “culture medium” because reference 10 cites another reference and I could not find specifics in that reference

·         In tables 1-3 under Risk factor, if it is blank, was the information not available?  Please make this clear to the reader. 

·         In figure 1, please clarify what “doubtful” histology results mean.

·         Minor: Line 348 #10 is repeated for this reference.

I recommend acceptance after minor edits.This is a retrospective evaluation of a blood-cell interferon-g release 29 assay (IGRA) with two different stage-specific mycobacterial antigens for the differential diagnosis of ADP suspected of mycobacterial origin.  This can be a major improvement since active/subclinical TB can be differentiated from latent TB by using these antigens.  There needs to be a great deal of further studies before it can be routinely used.  The time savings over waiting for cultures and the greater sensitivity over culture makes this important work. 

This study included a small sample size, N = 21, suspected to present TBLA without pulmonary involvement.  The authors broke the sample into three groups: N =8 cervical/axillary ADP, N = 10 mediastinal ADP, and N = 3 disseminated ADP. 

The authors were able to provide results for 15 patients.  The results of the combined HBHA and ESAT-6-interferon-gamma release assay were compared to culture.  This is a little problematic since culture is not truly a gold standard but this is all we have to compare results to.  Three of 6 had IGRA profile suggestive of active TB. Results are not noted for patients with alternative diagnosis determined.

Conclusion – HBH/ESAT-6 IGRA as a triage test has potential

There is some gaps in data, which are understandable, but the authors are cautious in their conclusions.

The authors address the limitation of the study. 

Recommendations from the reviewer:

·         In the methods section, the authors should add more details about the HBHA assay “culture medium” because reference 10 cites another reference and I could not find specifics in that reference

·         In tables 1-3 under Risk factor, if it is blank, was the information not available?  Please make this clear to the reader. 

·         In figure 1, please clarify what “doubtful” histology results mean.

·         Minor: Line 348 #10 is repeated for this reference.

I recommend acceptance after minor edits.This is a retrospective evaluation of a blood-cell interferon-g release 29 assay (IGRA) with two different stage-specific mycobacterial antigens for the differential diagnosis of ADP suspected of mycobacterial origin.  This can be a major improvement since active/subclinical TB can be differentiated from latent TB by using these antigens.  There needs to be a great deal of further studies before it can be routinely used.  The time savings over waiting for cultures and the greater sensitivity over culture makes this important work. 

This study included a small sample size, N = 21, suspected to present TBLA without pulmonary involvement.  The authors broke the sample into three groups: N =8 cervical/axillary ADP, N = 10 mediastinal ADP, and N = 3 disseminated ADP. 

The authors were able to provide results for 15 patients.  The results of the combined HBHA and ESAT-6-interferon-gamma release assay were compared to culture.  This is a little problematic since culture is not truly a gold standard but this is all we have to compare results to.  Three of 6 had IGRA profile suggestive of active TB. Results are not noted for patients with alternative diagnosis determined.

Conclusion – HBH/ESAT-6 IGRA as a triage test has potential

There is some gaps in data, which are understandable, but the authors are cautious in their conclusions.

The authors address the limitation of the study. 

Recommendations from the reviewer:

·         In the methods section, the authors should add more details about the HBHA assay “culture medium” because reference 10 cites another reference and I could not find specifics in that reference

·         In tables 1-3 under Risk factor, if it is blank, was the information not available?  Please make this clear to the reader. 

·         In figure 1, please clarify what “doubtful” histology results mean.

·         Minor: Line 348 #10 is repeated for this reference.

I recommend acceptance after minor edits.This is a retrospective evaluation of a blood-cell interferon-g release 29 assay (IGRA) with two different stage-specific mycobacterial antigens for the differential diagnosis of ADP suspected of mycobacterial origin.  This can be a major improvement since active/subclinical TB can be differentiated from latent TB by using these antigens.  There needs to be a great deal of further studies before it can be routinely used.  The time savings over waiting for cultures and the greater sensitivity over culture makes this important work. 

This study included a small sample size, N = 21, suspected to present TBLA without pulmonary involvement.  The authors broke the sample into three groups: N =8 cervical/axillary ADP, N = 10 mediastinal ADP, and N = 3 disseminated ADP. 

The authors were able to provide results for 15 patients.  The results of the combined HBHA and ESAT-6-interferon-gamma release assay were compared to culture.  This is a little problematic since culture is not truly a gold standard but this is all we have to compare results to.  Three of 6 had IGRA profile suggestive of active TB. Results are not noted for patients with alternative diagnosis determined.

Conclusion – HBH/ESAT-6 IGRA as a triage test has potential

There is some gaps in data, which are understandable, but the authors are cautious in their conclusions.

The authors address the limitation of the study. 

Recommendations from the reviewer:

·         In the methods section, the authors should add more details about the HBHA assay “culture medium” because reference 10 cites another reference and I could not find specifics in that reference

·         In tables 1-3 under Risk factor, if it is blank, was the information not available?  Please make this clear to the reader. 

·         In figure 1, please clarify what “doubtful” histology results mean.

·         Minor: Line 348 #10 is repeated for this reference.

I recommend acceptance after minor edits.

This is a retrospective evaluation of a blood-cell interferon-g release 29 assay (IGRA) with two different stage-specific mycobacterial antigens for the differential diagnosis of ADP suspected of mycobacterial origin.  This can be a major improvement since active/subclinical TB can be differentiated from latent TB by using these antigens.  There needs to be a great deal of further studies before it can be routinely used.  The time savings over waiting for cultures and the greater sensitivity over culture makes this important work. 

This study included a small sample size, N = 21, suspected to present TBLA without pulmonary involvement.  The authors broke the sample into three groups: N =8 cervical/axillary ADP, N = 10 mediastinal ADP, and N = 3 disseminated ADP. 

The authors were able to provide results for 15 patients.  The results of the combined HBHA and ESAT-6-interferon-gamma release assay were compared to culture.  This is a little problematic since culture is not truly a gold standard but this is all we have to compare results to.  Three of 6 had IGRA profile suggestive of active TB. Results are not noted for patients with alternative diagnosis determined.

Conclusion – HBH/ESAT-6 IGRA as a triage test has potential

There is some gaps in data, which are understandable, but the authors are cautious in their conclusions.

The authors address the limitation of the study. 

Recommendations from the reviewer:

·         In the methods section, the authors should add more details about the HBHA assay “culture medium” because reference 10 cites another reference and I could not find specifics in that reference

·         In tables 1-3 under Risk factor, if it is blank, was the information not available?  Please make this clear to the reader. 

·         In figure 1, please clarify what “doubtful” histology results mean.

·         Minor: Line 348 #10 is repeated for this reference.

I recommend acceptance after minor edits.

Author Response

Three of 6 had IGRA profile suggestive of active TB. Results are not noted for patients with alternative diagnosis determined

Response: Indeed, three of six patients with tuberculous cervical adenopathy had an IGRA profile suggestive of active TB. In contrast, the six patients with tuberculous mediastinal adenopathy and the three with disseminated tuberculous adenopathy had an IGRA profile suggestive of active TB.

Results for patients with alternative diagnosis are provided in Figure 1 and Table 4. 

In the methods section, the authors should add more details about the HBHA assay “culture medium” because reference 10 cites another reference and I could not find specifics in that reference

Response: Details about the culture medium and the antigen concentrations used have now been added in the methods section (lines 97-104).

In tables 1-3 under Risk factor, if it is blank, was the information not available?  Please make this clear to the reader.  

Response: This information was available for all the patients. When no risk factor was noticed, the symbol / was added in the table. The significance of this symbol has now been added in the table’s legend.

In figure 1, please clarify what “doubtful” histology results mean

Response: Histology was considered as positive (suggestive of TB) only when necrotic granuloma were noticed. Other abnormalities (epitheloïd granuloma, granuloma without necrosis, necrosis without granuloma) were considered as doubtful. This has now been defined in the figure legend.

Minor: Line 348 #10 is repeated for this reference.

Response: This has now been corrected for all references in the reference list.